# Male mating strategies to counter sexual conflict in spiders

Shichang Zhang [1,2], Long Yu [1,2], Min Tan [2], Noeleen Y. L. Tan [2,6], Xaven X. B. Wong [2,6], Matjaž Kuntner [1,3,4,5✉] & Daiqin Li [2✉]

When sexual conflict selects for reproductive strategies that only benefit one of the sexes, evolutionary arms races may ensue. Female sexual cannibalism is an extreme manifestation of sexual conflict. Here we test two male mating strategies aiming at countering sexual cannibalism in spiders. The "better charged palp" hypothesis predicts male selected use of the paired sexual organ (palp) containing more sperm for their first copulation. The "fast sperm transfer" hypothesis predicts accelerated insemination when cannibalism is high. Our comparative tests on five orbweb spider species with varying levels of female sexual cannibalism and sexual size dimorphism (SSD) reveal that males choose the palp with more sperm for the first copulation with cannibalistic females and that males transfer significantly more sperm if females are cannibalistic or when SSD is biased. By supporting the two hypotheses, these results provide credibility for male mating syndrome. They, however, open new questions, namely, how does a male differentiate sperm quantities between his palps? How does he perform palp choice after assessing his cannibalistic partner? By conducting follow-up experiments on *Nephilengys malabarensis*, we reveal that it is sperm volume detection, rather than left-right palp dominance, that plays prominently in male palp choice.

[1] State Key Laboratory of Biocatalysis and Enzyme Engineering & Centre for Behavioural Ecology & Evolution, School of Life Sciences, Hubei University, Wuhan 430062 Hubei, China. [2] Department of Biological Sciences, National University of Singapore, Singapore 117543, Singapore. [3] Department of Organisms and Ecosystems Research, National Institute of Biology, Ljubljana, Slovenia. [4] Jovan Hadži Institute of Biology, Scientific Research Centre of the Slovenian Academy of Sciences and Arts, Ljubljana, Slovenia. [5] Department of Entomology, National Museum of Natural History, Smithsonian Institution, Washington, DC, USA. [6] These authors contributed equally: Noeleen Y. L. Tan, Xaven X. B. Wong. ✉email: Matjaz.Kuntner@nib.si; dbslidq@nus.edu.sg

Sexual conflict occurs when males and females pursue their respective fitness optima at the expense of the other sex[1,2], and can culminate in antagonistic coevolution, a scenario in which adaptations in one sex select for counteradaptations in the other[3–6]. Also termed intersexual arms race, this evolutionary phenomenon is only sporadically documented, yet probably occurs widely in the animal kingdom[3,5–8]. Sexual cannibalism whereby a female kills and consumes a courting suitor[9], represents one of the most extreme manifestations of sexual conflict: in most cases it is not in the male's interest to be killed[10–12], but see ref. [13]. Sexual cannibalism should thus feature prominently in evolutionary research of arms races.

In spiders, preliminary reports of arms races encompass the evolution of extreme phenotypes[14]. Among them are sexual size dimorphism (SSD), male-biased sex ratios that underlie high sperm competition, polyandry, monogyny, genital mutilation, and sexual cannibalism[11,12,14–21]. Sexual cannibalism is widespread in sexually dimorphic spider lineages[12,14,18,22]. It may have slightly different functions if it occurs prior to, or after copulation, but if successful, sexual cannibalism precludes male mating or re-mating. Within a classical arms race scenario that we here term *the male mating syndrome*, male spiders may aim at countering cannibalism's zero remating effect with physiological and/or behavioural counteradaptive mating strategies. The proximate physiological mechanisms may include sperm activation and the role of seminal fluids[23]. Examples of behavioural strategies that augment male reproductive success and paternity in spite of cannibalism seem to be more numerous. These include mate guarding, mate binding, opportunistic mating, sacrificial death during copulation, and genital mutilation, even emasculation[12,14–16,18,19,22,24,25].

We explored whether male spiders exert additional cannibalism countering strategies. By formulating two testable hypotheses explained below, we tested the validity of the *male mating syndrome*. Within a cannibalistic mating regime, males may face a single shot at mating, but since all male spiders have paired sexual organs (palps), we expected their preferential use of the palp that was better charged with sperm[25]. The use of a better charged palp would grant the male a greater potential in transferring more sperm into the female at a given copulation duration, thereby increasing paternity in a system of polyandrous females and high levels of sperm competition. It has only been recently documented that spider palps are innervated[26–28]. Theoretically then, a mate-seeking male is able to sense the uptake and release of sperm in his paired organs, which would facilitate palp use choice. Since it is unknown whether this hypothetical male strategy may be related to female cannibalism levels, we tested it here as the "better charged palp" hypothesis. It predicts preferential use of the better charged organ with a female that is cannibalistic.

Plausibly, increasing the rate of sperm transfer may be another critical step in increasing a male's fitness. This "fast sperm transfer" hypothesis posits that, in addition to using a palp with more sperm, a male that faces cannibalism attempts may also benefit in fitness by increasing the rate of sperm transfer. This hypothesis predicts increased sperm transfer when mating with a cannibalistic female as it would reduce the time exposed to a cannibal, while achieving sufficient insemination. These two hypotheses are not mutually exclusive.

In the comparative part of this paper, we tested the two hypotheses by studying male palp choice and rate of sperm transfer in relation to female sexual cannibalism and SSD in five orbweb spider species from two families: *Argiope versicolor* (Doleschall, 1859), *Herennia multipuncta* (Doleschall, 1859), *Nephila pilipes* (Fabricius, 1793), and *Nephilengys malabarensis* (Walckenaer, 1841) (Araneidae), and *Leucauge decorata*

(Blackwall, 1864) (Tetragnathidae) (Classification follows[29], but for the alternative, see ref. [30]). We chose these five taxa to represent gradients in sexual cannibalism and SSD. We hypothesised that the two male mating strategies (palp choice and sperm transfer rate adjustment) vary with the levels of female sexual cannibalism and/or SSD. Indeed, our comparative study gave credibility to the male mating syndrome through experimental tests that supported the "better charged palp" and the "fast sperm transfer" hypotheses. However, this opened the questions of how male spiders recognise sperm quantities in their palps, and how they perform palp choice after assessing if a female is cannibalistic. We proposed the mechanism hypotheses according to two ways a male spider may fill up its palps with sperm. First, mechanism A assumes that a random amount of sperm may be loaded into each palp and the spider is then able to detect minute differences in sperm quantity of each palp to make a choice of palp during copulation. Second, mechanism B assumes that the male may intentionally load one palp with more sperm than the other and thus prefer to use the palp containing more sperm. In this latter case, memory may be involved in palp selection (for details see "Methods"). We tested these male genital choice mechanisms to better understand the biology of the male mating syndrome by studying *N. malabarensis* through a series of follow-up experiments.

Our results from comparative tests on five orbweb spider species with varying levels of female sexual cannibalism and SSD showed that males choose the palp with more sperm for the first copulation with cannibalistic females and that males transfer significantly more sperm if females are cannibalistic or when SSD was highly biased. Our follow-up experiments focusing on *N. malabarensis* revealed that it is sperm volume detection, rather than left-right palp dominance, that plays prominently in male palp choice.

## Results

**Five spider species with varying levels of female cannibalism**. In total, we conducted 86 successful mating trials across five species (*A. versicolor* [AV]: $N = 18$; *H. multipuncta* [HM]: $N = 11$; *L. decorata* [LD]: $N = 17$; *N. malabarensis* [NM]: $N = 28$; *N. pilipes* [NP]: $N = 12$) (see Supplementary Figs. 1–5). We found that the rate of sexual cannibalism varied among the five species (Supplementary Table 1). Here, we defined an act of sexual cannibalism as when the female grabbed and/or killed her mate before, during or after a male's first insertion (Supplementary Fig. 6). According to this definition, *A. versicolor* was the most sexually cannibalistic species: aggressive females usually killed the males during the process of first copulation (26.3%). This was followed by *N. malabarensis* (21.4%), *L. decorata* (11.8%), and *H. multipuncta* (9.1%). The least sexually cannibalistic species was *N. pilipes* (0%): while females showed the most aggressiveness towards males, they did not attack and kill them.

**No significant difference in sperm quantities between the left and right palps**. For each mating trial, we noted which palp (left or right) was used first for insertion (palp choice). Among the five species, full palp breakage only occurred in *N. malabarensis* (Supplementary Figs. 7–9) and we immediately removed any broken palp (Supplementary Fig. 7) to prevent continuous sperm transfer. To test whether palp choice was biased towards the left or right palp, we compared the mean sperm count from the left and right palp of the five species. For each species, we performed paired *t*-test to test for the difference in the mean number of sperm between the left and right palp. We found no significant difference in sperm count between the pair of palps for each of the five species (Fig. 1a and Table 1). This suggests no prior bias in sperm charging for either palp. For each species, we conducted

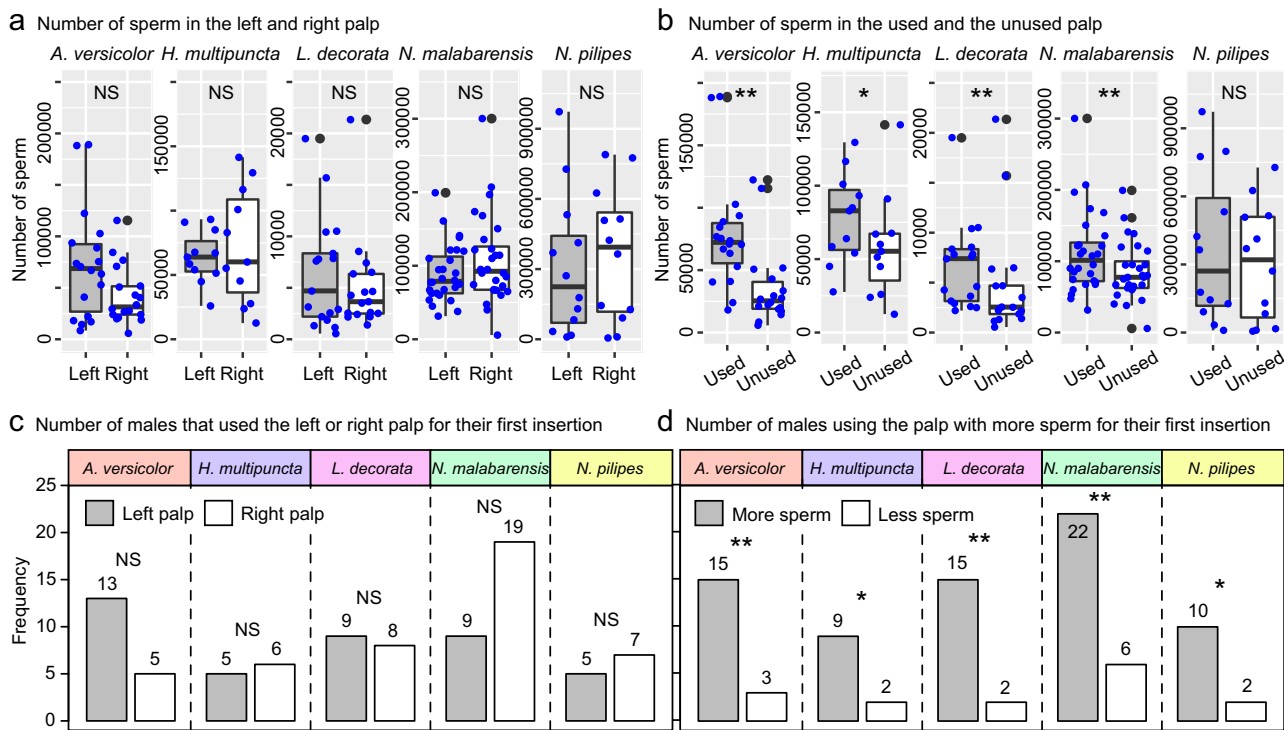

**Fig. 1 Number of sperm in male palps and frequency of palp use during the first insertion. a** There was no significant difference in the number of sperm between the left and right palp in all five species. **b** The number of sperm was higher in the palp used in the first insertion for four species. **c** Males for all five species had no preference for the left or right palp in their first insertion. **d** For their first insertion, males chose to use the palp with more sperm for all five species. Boxplots show the median (central line), first and third quartiles (box), and minimum and maximum values (whiskers). NS: not significant, *: $p < 0.05$, **: $p < 0.01$ (for statistical results, see Table 1).

---

**Table 1 Comparisons of the mean (± s.e.m.) number of sperm between the left and right palp, between the used and unused palp for the first insertion, as well as between the number of males that used the left or right palp for the first copulation and between the number of males that used the palp with more or less sperm.**

| | | A. versicolor | H. multipuncta | L. decorata | N. malabarensis | N. pilipes |
|---|---|---|---|---|---|---|
| No. of sperm in left and right palps | Left | 72,683 ± 12,608 | 63,864 ± 6,149 | 6,356 ± 1,310 | 89,380 ± 7,363 | 311,342 ± 88,449 |
| | Right | 42,458 ± 6,644 | 72,364 ± 13,493 | 5,094 ± 1,131 | 107,720 ± 11,399 | 364,867 ± 82,471 |
| | Paired $t$ test | $t_{17} = 1.75$ | $t_{10} = -0.27$ | $t_{16} = 0.80$, | $t_{27} = -1.58$ | $t_{11} = -0.86$ |
| | N | 18 | 11 | 17 | 28 | 12 |
| | p | 0.098 | 0.796 | 0.436 | 0.126 | 0.410 |
| No. of sperm in used and unused palps | Used | 78,925 ± 10,826 | 77,950 ± 9,411 | 6,694 ± 1,056 | 111,673 ± 10,864 | 370,738 ± 95,674 |
| | Unused | 36,217 ± 7,690 | 58,277 ± 10,751 | 4,756 ± 1,347 | 85,427 ± 7,719 | 306,221 ± 73,988 |
| | Paired $t$ test | $t_{17} = 3.68$ | $t_{10} = 2.41$ | $Z = -2.77$ | $t_{27} = 3.13$ | $t_{11} = 0.92$ |
| | N | 18 | 11 | 17 | 28 | 12 |
| | p | 0.002 | 0.037 | 0.006 | 0.004 | 0.379 |
| No. of males used left or right palp | Left | 13 | 5 | 9 | 9 | 5 |
| | Right | 5 | 6 | 8 | 19 | 7 |
| | Chi-square test | $\chi^2_1 = 3.56$ | $\chi^2_1 = 0.09$ | $\chi^2_1 = 0.06$ | $\chi^2_1 = 3.57$ | $\chi^2_1 = 0.33$ |
| | N | 18 | 11 | 17 | 28 | 12 |
| | p | 0.059 | 0.763 | 0.808 | 0.059 | 0.564 |
| No. of males used the palp with more or less sperm | More | 15 | 9 | 15 | 22 | 10 |
| | Less | 3 | 2 | 2 | 6 | 2 |
| | Chi-square test | $\chi^2_1 = 8.00$ | $\chi^2_1 = 4.46$ | $\chi^2_1 = 9.94$ | $\chi^2_1 = 9.14$ | $\chi^2_1 = 5.33$ |
| | N | 18 | 11 | 17 | 28 | 12 |
| | p | 0.005 | 0.035 | 0.002 | 0.002 | 0.021 |

separate Chi-square tests for goodness-of-fit to check if males exhibited a preference to use the left or right palp. While *A. versicolor* and *N. malabarensis* males tended to use the left and right palp, respectively, during the first insertion, these differences were not statistically significant (Fig. 1c and Table 1).

**Males used the palp with more sperm for their first insertion**. To test the prediction that the first palp used for insertion would contain more sperm than the unused palp, we compared the mean sperm count from the used and unused palp of the five species. For the four species with a normal distribution (*A. versicolor, H. multipuncta, N. malabarensis, N. pilipes*), we performed a paired *t*-test to test if there was a significant difference in the mean number of sperm between the used and unused palp. As *L. decorata* had a non-normal distribution, we conducted a Wilcoxon signed-rank test instead. With the exception of *N. pilipes*, the palp used for the first insertion contained significantly

more sperm than the unused palp (Fig. 1b and Table 1). Furthermore, to test how often male spiders chose to use the palps with more sperm for each species, we performed separate Chi-square tests for goodness-of-fit for each species. Significantly more males used the palp with more sperm for their first insertion across all five species (Fig. 1d and Table 1).

**The "better charged palp" hypothesis**. To test this hypothesis, we conducted sperm counting, and analysed the data using generalised linear models (GLMs) with a Gamma error and log-link function (Supplementary Table 2). The effects of sexual cannibalism (whether or not a female was cannibalistic; coded as 1 or 0 for each female), sexual size dimorphism (SSD; ratio of female mass to male mass), male post-maturation age (male age), female post-maturation age (female age), species, and taxonomy (2 families) were included as fixed factors. We also included 3 two-way interactions of sexual cannibalism with SSD, male age and female age in the models. According to Weber's law (see "Methods"), a male's ability to discriminate the difference of sperm between its two palps based on their ratio or proportional difference, rather than on the absolute difference between them. We included the ratio of sperm count in the used to the unused palp as the response variable We proposed a series of models by including different number of explorative variables and then compared the models using Akaike's information criterion corrected for small sample sizes (AICc) to identify the best fitting model that predicts the effects of explorative factors on palp choice (for statistical analyses, see "Methods").

We found that the model including the six predictors alone, and interactions of sexual cannibalism with female and male post-maturity age, was the best fitting model (Table 2). However, only species and the two aforementioned interactions had significant effects on how often the males used the palp with more sperm for the first insertion (Table 2). While males significantly chose the palp with more sperm for their first insertion in all five species (Fig. 1d), *A. versicolor* males chose the palp with more sperm more frequently compared to *H. multipuncta* and *L. decorata* (Table 2). For two significant interaction terms, sexual cannibalism affecting how frequently the males used the palp with more sperm for the first insertion depended on female and male post-maturity age. Males were more likely to use the palp with more sperm when mating with older and cannibalistic females ($\beta = 0.02$, $t = 2.76$, $p = 0.008$; Fig. 2a) or

**Table 2 The best fitting GLM model including six predictors alone (taxonomy, species, presence of sexual cannibalism, sexual size dimorphism, female age, male age) and two two-way interactions to predict whether the male selects the palp with more sperm.**

| Exploratory factors | β | s.e. | t | p |
|---|---|---|---|---|
| Intercept | 0.87 | 0.26 | | |
| Taxonomy (Tetragnathidae) | −0.14 | 0.26 | −0.54 | 0.594 |
| Species (HM) | −0.89 | 0.25 | −3.50 | 0.001 |
| Species (NM) | −0.50 | 0.29 | −1.73 | 0.088 |
| Species (NP) | −0.61 | 0.28 | −2.18 | 0.033 |
| Species (LD) | NA | NA | NA | NA |
| Female cannibalism | 0.44 | 0.38 | 1.15 | 0.253 |
| Sexual size dimorphism | 0.003 | 0.003 | 0.88 | 0.383 |
| Female age | 0.004 | 0.004 | 0.89 | 0.378 |
| Male age | −0.001 | 0.008 | −0.16 | 0.875 |
| Female cannibalism: Female age | 0.02 | 0.01 | 2.76 | 0.008 |
| Female cannibalism: Male age | −0.04 | 0.01 | −3.17 | 0.002 |

AV: *A. versicolor*, N = 18; HM: *H. multipuncta*, N = 11; LD: *L. decorate*, N = 17; NM: *N. malabarensis*, N = 28; NP: *N. pilipes*, N = 12.

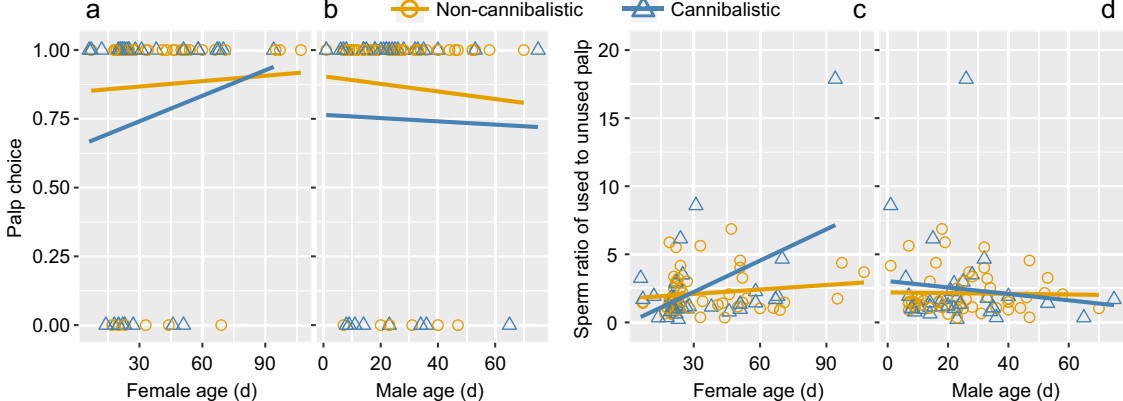

**Fig. 2 Effect of the interactions of sexual cannibalism with female post-maturity age and with male post-maturity age on how frequently males chose the palp with more sperm for their first insertion. a** Significantly more males chose the palp with more sperm when mating with older and cannibalistic females (blue line). **b** Younger males chose the palp with more sperm when mating with non-cannibalistic females (orange line). **c** Males selected the palp with a significantly higher sperm ratio when mating with an older female only if she was cannibalistic (blue line). **d** Younger males selected the palp with a significantly higher sperm ratio with a cannibalistic partner (blue line).

when the males were young and mated with non-cannibalistic females ($\beta = -0.04$, $t = -3.17$, $p = 0.002$; Fig. 2b). A similar trend was observed for the sperm ratio (Fig. 2c, d). Whether a female was cannibalistic was weakly positively correlated with female age (Kendall's rank correlation: $r = 0.017$, $z = 2.15$, $p = 0.032$), but

was not correlated with male age ($r = 0.195$, $z = 0.19$, $p = 0.851$). All other factors (i.e., taxonomy, sexual cannibalism, SSD, female age and male age) alone had no significant effects on palp choice (Table 2).

**The "fast sperm transfer" hypothesis.** To test for the effects of the six aforementioned explorative variables on the percentage of sperm transferred to female spermathecae (i.e., number of sperm transferred / total number of sperm in the used palp × 100%), we ran a series of GLMs with a Gamma error and log-link function by including different explorative variables in the models (Supplementary Table 3). The best fitting model showed that taxonomy, species, sexual cannibalism and SSD alone best predicted the percentage of sperm transferred into the female's spermathecae (Table 3). Compared to the araneid *A. versicolor*, males of the tetragnathid *L. decorata* transferred more sperm during first insertion ($\beta = 0.60$, $z = 2.09$, $p = 0.036$). The mean percentage of sperm transfer ranged from 20 to 40% for most species (Fig. 3a). *Nephila pilipes* males transferred up to 95% sperm during their copulation. Sexual cannibalism alone significantly affected the percentage of sperm transfer ($\beta = 0.94$, $z = 2.16$, $p = 0.031$): males generally transferred more sperm when females were cannibalistic (Fig. 3b). This was also apparent for all species with the exception of *N. malabarensis* (Fig. 3a). SSD alone also had a significant effect on the percentage of sperm transfer ($\beta = 0.01$, $z = 2.97$, $p = 0.003$): the more biased SSD was, the more sperm were transferred (Fig. 3c). This trend was observed for all species except *A. versicolor*, where greater SSD was associated with lower percentage of sperm transferred (Fig. 3d). Female age, male age, copulation duration and all the interactions had no significant effects on percentage of sperm transfer (Supplementary Table 3).

**Table 3 The best fitting GLM model including seven predictors alone (family, species, presence of sexual cannibalism, copulation duction, sexual size dimorphism, female age, male age) and four two-way interactions to predict the percentage of sperm transferred to the female spermatheca.**

| Exploratory factors | β | s.e. | z | p |
|---|---|---|---|---|
| Intercept | 2.78 | 0.28 | | |
| Family (Tetragnathidae) | 0.60 | 0.28 | 2.09 | 0.036 |
| Species (HM) | −0.10 | 0.29 | 0.33 | 0.738 |
| Species (NM) | −0.60 | 0.30 | 2.00 | 0.046 |
| Species (NP) | −1.38 | 0.50 | 2.74 | 0.006 |
| Species (LD) | NA | NA | NA | NA |
| Female cannibalism | 0.94 | 0.43 | 2.16 | 0.031 |
| Sexual size dimorphism | 0.01 | 0.004 | 2.97 | 0.003 |
| Female age | −0.001 | 0.005 | 0.23 | 0.815 |
| Male age | −0.005 | 0.009 | 0.54 | 0.592 |
| Copulation duration | 0.00003 | 0.001 | 0.04 | 0.970 |
| Female cannibalism: Sexual size dimorphism | 0.01 | 0.01 | 1.66 | 0.097 |
| Female cannibalism: Female age | −0.02 | 0.01 | 1.78 | 0.075 |
| Female cannibalism: Male age | −0.02 | 0.01 | 1.45 | 0.147 |
| Female cannibalism: Copulation duration | 0.001 | 0.001 | 1.20 | 0.230 |

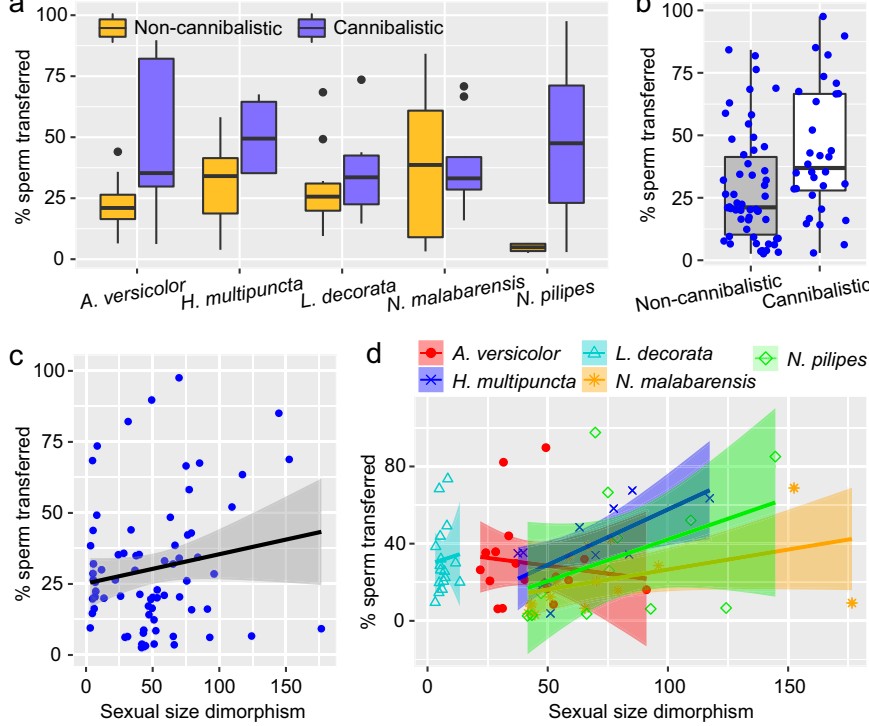

**Fig. 3 Effects of species, sexual cannibalism and sexual size dimorphism on the percentage of sperm transferred. a** The percentage of sperm transferred when a male mated with a cannibalistic or non-cannibalistic female in the five species. **b** Overall, males transferred significantly more sperm if a female was cannibalistic. **c** A positive correlation between sexual size dimorphism and the percentage of sperm transferred across the five species. **d** The more biased sexual size dimorphism, the more sperm were transferred in all species except in *Argiope versicolor*.

**Table 4 The best fitting GLMM model including tree fixed effects (SSD, positive difference in sperm number, sperm ratio and sexual size dimorphism, and palp breakage) and one random effect (male identity) to predict the palp choice.**

| Exploratory factors | $\beta$ | s.e. | t | p |
|---|---|---|---|---|
| Intercept | 5.60 | 2.30 | | |
| Log(positive difference) | −0.02 | 0.14 | −0.15 | 0.884 |
| Log(sperm ratio) | −0.25 | 0.64 | −0.38 | 0.708 |
| Log(SSD) | −1.09 | 0.44 | −2.50 | 0.049 |
| Palp breakage | −0.27 | 0.40 | −0.66 | 0.523 |

**Mechanisms of palp choice**. To elucidate how a male spider recognises sperm quantity differences between its paired sexual organ (palp) and performs palp choice after assessing if a female is cannibalistic, we performed a series of mating experiments on *N. malabarensis*. Unlike the mating trials performed in the previous sections, where males were allowed to copulate once and female cannibalism was prevented, we here allowed the males to perform multiple insertions with multiple females for up to five hours or until palp mutilation occurred. We designed these multiple mating trials such that each successful copulation with a different female would change the amount of sperm in the used palp. However, we prevented continuous sperm induction after the first insertion by sealing the males' genital pores with a thin layer of super glue (Selleys, PTY Ltd, Australia) using fine brush[24]. We counted sperm in the used palp after each insertion for males and in the female spermathecae after each copulation. In total, 34 copulations were performed by 23 males. Eight males copulated more than once, with two to four copulations conducted during 13 subsequent mating trials. The remaining 15 males copulated once.

Our results showed a significant difference in the amount of sperm for the palps that were used (209,842 ± 22,367) and not used (163,077 ± 18,141) during copulation in 13 subsequent mating trials only (Wilcoxon signed rank test: $Z = -2.341$, $N = 13$, $p = 0.019$). The amount of sperm in the used palp was significantly greater than that of the unused palp. Out of 13 subsequent mating trials, males in 10 trials chose the palp with more sperm for their first copulation, but this was only marginally significant (Chi-square test for goodness-of-fit: $\chi^2 = 3.77$, $df = 1$, $p = 0.052$).

The amount of sperm in the unused (172,888 ± 12,801) and used (197,315 ± 13,058) palps for all 34 copulations was not significantly different (paired *t*-test: $t_{33} = -1.744$, $N = 34$, $p = 0.091$). Nonetheless, the amount of sperm in the used palp was generally greater than that of the unused palp. Males chose the palp containing more sperm for copulation significantly more frequently than they chose the palp with less sperm regardless of trials (23 out of 34 copulations: $\chi^2 = 4.24$, $df = 1$, $p = 0.040$). Thus, even non-virgin males still tended to select the palp with more sperm for their subsequent copulations.

Out of the eight males that copulated more than once, three consistently used the same palp for copulation in the subsequent mating trials. However, this difference was not statistically significant ($\chi^2 = 5.00$, $df = 1$, $p = 0.48$). This suggests that palp choice may not be made based on palp dominance (i.e., if they preferred to use the left or right palp for copulation in most circumstances analogous to the left- or right-handedness in humans) since there was no strong bias towards the use of one particular palp.

To elucidate how a male recognises the difference in sperm quantities between its two palps and cognitively chooses the better charged palp, we performed a series of generalised mixed-effects models with binomial error and logit-link function. We included the positive difference in sperm amount between two palps, sperm ratio (i.e., the ratio of the amount of sperm present in the palp with more sperm to that in the palp with less sperm), SSD and emasculation frequency as fixed effects and male identity as a random effect. While the full model containing all the fixed effects was the best fitting model (Supplementary Table 4), all the factors had no significant effects on palp choice except that SSD had a marginal effect (Table 4).

We then set a cut-off point at the 75th percentile of positive difference in sperm amount and sperm ratio of the mating trials where males did not choose the palp with more sperm. Thus, small and large positive differences or low and high sperm ratios were distinguished. The 75th percentile of the positive difference and sperm ratio for the situations where the palp with less sperm was chosen was set at 95,300 and 1.634, respectively.

We then carried out four separate Chi-square tests for goodness-of-fit. (1) In mating trials whereby males had a positive difference of sperm >95,300 prior to copulation, significantly more males (9 out of 11) selected the palp with more sperm ($\chi^2 = 4.455$, $df = 1$, $p = 0.035$). (2) In mating trials whereby males had a positive difference of sperm which was ≤95,300, more males (14 out of 23) selected the palp with more sperm, but the difference was not statistically significant ($\chi^2 = 1.087$, $df = 1$, $p = 0.297$). This indicates that palp choice was more random or they could not discriminate the difference in sperm amount between two palps when positive difference ≤95,300. (3) In mating trials that included those males with the ratio of the amount of sperm present in the palp with more sperm to that in the palp with less sperm > 1.634, significantly more males (10 out of 12) selected the palp with more sperm ($\chi^2 = 5.333$, $df = 1$, $p = 0.021$). (4) In mating trials with those males with a sperm ratio ≤1.634, more males (13 out of 22) selected the palp with more sperm, but the difference was not statistically significant ($\chi^2 = 0.727$, $df = 1$, $p = 0.394$). Here too, the palp choice was more random when sperm count ratio ≤1.634.

Taken together, these results suggest that when the absolute difference in sperm quantity or the ratio between two palps was below a threshold (95,000 in sperm amount or 1.634 in sperm ratio), males were unable to differentiate between their palps. This was more likely to occur in the repeated mating trials where sperm was depleted and palp use was alternated.

## Discussion

Male spiders are able to assess a female's cannibalism level before choosing which palp to use for copulation. Our comparative tests on five orbweb spider species show (1) that when a female is cannibalistic and older or when a male is younger, the male is more likely to choose the palp with more sperm; (2) a male transfers more sperm into the female if she is cannibalistic and if SSD is biased, regardless of copulation duration, male age and female age. These results support the "better charged palp" and the "fast sperm transfer" hypotheses. Our results are in line with the sexual conflict theory whereby sex-specific optimal fitness is pursued at the expense of the other sex. A part of this theory is our generally circumscribed idea of the male mating syndrome: male spiders are under sexual conflict pressure in those systems that are highly sexually cannibalistic, as they may only have a single chance to mate. The outcomes of our comparative tests warranted immediate follow-up experimental work intended to elucidate the ability of males to recognise the difference in sperm quantity between their paired sexual organs, as well as mechanisms of their cognitive ability to choose the one palp with more sperm.

Our key finding is that, for all five species, a male is more likely to choose the palp with more sperm for the first copulation. By

detecting no significant difference in how often males use their left or right palp for the first insertion and also no significant difference in the number of sperm between the left and right palp for all five species, these results rule out the possibility of a bias towards the left, or right palp. Critically, males choose the palp with more sperm more frequently when females are more cannibalistic and older, or when young males mate with cannibalistic females. Considering our definition of cannibalism (when female grabs or kills her suitor), mating with a cannibal provides only a single mating opportunity for the male, particularly when mating with older females that are shown to be more cannibalistic. A single mating chance in a lifetime often triggers mechanisms commonly referred to as terminal investment behaviour, whereby a male invests all his remaining resources at once[22,31–33], including all available sperm. Therefore, the strategy to use the palp with more sperm enhances his potential of transferring more sperm, depending on copulation duration and the rate of sperm transfer. A male with zero chances for subsequent copulations will benefit from transferring as many sperm as possible thereby maximising resource investment with—most likely—the only female partner[34].

Choosing the palp with more sperm is logically expected in species with palpal mutilation, a common mating strategy in monogynous males facing sexually cannibalistic females (e.g., Araneidae and Theridiidae[19,35,36]). Within our comparative sample, only *N. malabarensis* males engage in full emasculation, i.e., severing one or both palps entirely ("half-eunuchs" and "eunuchs", respectively[17,18]). Unlike full eunuchs, half-eunuchs may have another potential shot at mating if they successfully escaped a cannibalistic female. Furthermore, "remote copulation" biology, which describes a mechanism of continuous sperm transfer from a severed organ into the female, is known in *N. malabarensis*[37]. This means that, despite being cannibalised, male eunuchs can benefit from selecting the better charged palp thereby maximising sperm transfer. Although the other four species studied here do not engage in full emasculation, lesser genital mutilation may facultatively occur in *A. versicolor, H. multipuncta* and *N. pilipes*[14,15,19,35,36]. Since damaged organs may malfunction as sperm transferring devices after their first use, partially damaged males may likewise benefit from preferential use of the better charged palp, analogous to eunuchs.

However, our results detect a pattern that is difficult to explain. Why do younger males select the palp with more sperm more frequently when facing non-cannibalistic females, but are more likely to select the palp with higher ratio (ratio of sperm amount) if a female is cannibalistic? Patterns of genital damage depending on male and female age, as well as levels of cannibalism may be at work here, as may be as yet enigmatic patterns of sperm precedence in polyandrous spider systems. Clearly, additional experiments are needed to elucidate these questions.

We show that sexual cannibalism alone significantly affects the amount of sperm transferred: males transferred sperm more quickly and in greater quantities when mating with cannibalistic females. In addition, males transferred more sperm to females when SSD was more biased: males have a higher chance of being cannibalised when facing a larger female in a sexual size dimorphic spider system. The speed of sperm transfer, therefore, matters to the male.

Our results suggest that males are able to assess a female for the danger of cannibalism, then use the palp with more sperm during copulation. We thus followed up with additional experiments to elucidate the mechanisms of the male's cognitive ability to differentiate between sperm quantities in each palp.

A cognitive choice of palp for copulation would be refuted if males were to choose their palp based on left or right palp dominance. In our second set of experiments, the amount of sperm in the male's palps was allowed to be reduced over the course of several mating trials. If the male spider consistently used one palp throughout the mating trials regardless of the amount of sperm in each palp, it is possible that palp dominance influences palp choice. However, we did not observe any significant preference for palp choice. Instead, most males alternated their choice of the palp in subsequent trials. Although sample sizes were limited ($N = 8$) due to the emasculation behaviour, our results refute the role of palp dominance.

It is plausible that during sperm induction—a process unique to spiders during which a male ejaculates from gonopore onto sperm web and then sucks the sperm into his palps—the male will intentionally charge one palp more than the other. Hypothetically, this scenario would necessitate the male's remembrance of which palp is the better charged one. However, it seems more likely that males are able to recognise the difference in sperm quantity between the palps.

Weber's law describes the ability of a sensory system to discriminate between two stimulus magnitudes based on their ratio or proportional difference, rather than on the absolute difference between them[38–41]. As the ratio becomes smaller, comparisons become more difficult. When sensory systems adhere to Weber's law, the smallest change that can be detected is a constant proportion of the stimulus magnitude. For example, if a male can detect the proportional difference of one-tenth its standard length, then a 10 mm spider can detect at least 1 mm of size difference between itself and its opponent. On the other hand, a 50 mm spider could only detect 5 mm of size difference, and any male that differs in size by less than 5 mm would not be identified as being 'smaller' or 'larger'.

We predicted that, in accordance with Weber's law, only males with palps differing in sperm quantity above a certain discrimination threshold can discriminate which palp is more or less charged with sperm. Indeed, our results show that palp choice was random below a certain discrimination threshold (i.e., 93,500 in number of sperm or 1.634 in the sperm ratio between the palps). On the other hand, males were able to discriminate between sperm quantity or ratio above that threshold. This strongly suggests that proportional processing may influence a male spider's palp choice decision, which may be based on magnitude-dependent detection of change[38,42], though the sensory systems involved remain enigmatic. In addition, sensory systems might evolve such that it is the discrimination thresholds, not the male mating strategies, that change. The role of proportional processing may be critical to understand sexual conflict.

Sexual cannibalism is shown to be the driving force for the male counter adaptive strategies: palp choice and sperm transfer speed. Interestingly, other factors such as SSD, male and female age, as well as species, also significantly influenced them. As such, future studies should focus on a single strategy and look at more explorative factors (such as body condition, and details of genital damage) to identify the most important factor(s) behind the adaptation. Although sexual cannibalism frequencies in past studies on the five genera drastically differ (*Argiope*: 36–80%[43–47]; *Herennia*: 50%[15]; *Leucauge*: 9–30%[48]; *Nephila pilipes*: 17%[24]; *Nephilengys*: 75%[17]) in our study, variation in sexual cannibalism across species only ranged from 0 to 26.32%. This difference in cannibalism frequency may not be significant enough to determine a male mating strategy. Hence, future studies should study sexual cannibalism frequencies over a broader taxonomic sample, perhaps varying the strictness of cannibalism definition, and its timing.

As we strived to understand the copulatory behaviour of male spiders in this study, future research is encouraged to better understand the rationale behind male palp choice. Assuming that volume detection is the fundamental mechanism for palp

selection in orb-weaving spiders, the anatomical, neuronal and hormonal aspects of male palp choice may be explored to further understand how a male spider distinguishes relatively minute differences in sperm volumes. Furthermore, the large diversity of extant spiders globally indicates that these findings may not apply to all sexually dimorphic and cannibalistic spiders. Due to sexual selection, the palpal anatomy can undergo rapid changes, such that mechanisms behind sperm transfer of various species may differ. In conclusion, this study supports the male mating syndrome within a sexual conflict paradigm.

## Methods

**Study species and spider maintenance.** *Argiope versicolor*, *H. multipuncta*, *L. decorata*, *N. pilipes* and *N. malabarensis* are commonly found in the subtropics (Supplementary Figs. 1–5) and tropics. With the exception of *Leucauge*, all abovementioned four genera are currently catalogued in the family Araneidae[30]. We collected large juveniles and subadults of the five species in Singapore.

Sexual cannibalism in the genus *Argiope* takes place before and during copulation[44,46,49,50]. The cannibalism frequency is reported to be around 36–80%, regardless of the female mating status[43,46,47]. Courtship begins when the male cuts into the female web to which he connects a mating thread (Supplementary Fig. 1); the latter forms a substrate for vibrational courtship. The female moves from her position on the web to the mating thread, where the male moves down to copulate. Sometimes, males are eaten during the first copulation trial. If the male is not cannibalised, he will repeat his courtship for a second insertion. However, the highly cannibalistic *Argiope* often sees the males getting cannibalised within the first two insertions[43,50]. This would render the copulation duration to be very short, about 8 s on average[47]. Despite the short copulation duration, the fertilisation rate and hatching success are not correlated[47]. If this was so, the possible reason for males attempting a second insertion could be to increase the number of sperm transferred for success in sperm competition[47,51].

Courtship and mating in *H. multipuncta* occur on the female's web[52]. Several adult males can live on the web of a female, even if the female is immature[53]. The epignum of the female is snugly blocked as she positions herself on the web (Supplementary Fig. 2), which means that when a male tries to copulate with her, she will, if receptive, raise her body out of the 'hub-cup' for access to her epigynum[53]. This may, however, be the initiating point of her aggressiveness and cannibalism if a male tries to squeeze into her 'hub-cup' (personal observations). Post-copulatory palp breakage may serve as a mating plug, and it is observed in aggressive females exhibiting sexual cannibalism, which occurs 50% of time[15].

According to Aisenberg (2009), *L. mariana* was observed not to be sexually cannibalistic[54], although a more recent study reports its cannibalism frequency to be 9–30%[48]. Although the *Leucauge* species used in this study is different, it is a reasonable prediction that its cannibalism levels are similar. Among the five studied species, *L. decorata* exhibits the lowest SSD (Supplementary Fig. 3). Females appear to form their own mating plugs by secreting a substance after copulation[55,56], which may hinder future copulation. During courtship, the pair uses chelicerae clasping to hold onto each other while the male stretches out his palp for insertion[21] (Supplementary Fig. 3). Each insertion appears to be in a form of short and long tappings and occur multiple times, where sperm transfer occurs[54]. Males also use their legs to beat the females' abdomen which visibly acts as an enhancement for copulation approval by the female[54]. If the female is receptive, she will position herself to face the male for the copulation to start.

SSD is the most pronounced in *N. pilipes* (Supplementary Fig. 4), which could be the possible reason for females failing to detect approaching males or ignoring them since male size is too small for female optimal foraging[57]. Sexual cannibalism is observed before and during copulation, with the frequency observed to be at around 17% in *N. pilipes*[24]. Like *Herennia*, several males are observed to be on the web of the female[57–59]. Courtship by the male includes a unique mate binding behaviour where it spins silk on the female's abdomen, legs and carapace[24,51].

Females are highly cannibalistic in *N. malabarensis*[17] with sexual cannibalism levels up to 75%. Because of this, male *Nephilengys* often break off their palps during copulation (Supplementary Fig. 5) to avoid being cannibalised while continuing to transfer sperm into the female's spermatheca to increase male chances of producing progeny[37]. Eunuch males also benefit from being lighter and agile: the 'gloves-off' hypothesis predicts enhancement of eunuch survival through higher physical stamina and endurance[17]. Since the advantages of palp severance appear to exceed its cost, most males break off their palps.

We maintained spiders in the controlled environmental conditions following the standard protocols used in earlier studies[24,37,60]. We reared and monitored them until they matured. We used only virgins (both males and females) in this study and recorded their post-maturity ages from the last moult, which aided in standardising the ages of female spiders used in the experiment as an attempt to eliminate age as a confounding factor. A total of 86 successful mating trials were conducted for this study. The mating trial procedures are summarised below.

**Pre-copulation procedures.** To control for hunger possibly affecting sexual cannibalism and palp choice, we fed the test spiders 24 h before the trials. An hour prior to the mating trials, we measured the spiders' mass, total body length and carapace width, and then returned them back to their housing cages.

**Mating trials.** As males reach their partners on female webs to perform ritualised vibratory courtships[61], we staged all mating trials in the resident female cages. We started a mating trial by releasing a male at the bottom of the cage, at a minimum distance of 10 cm to the female. We encouraged the test male to continue to approach the female every 10 min gently using a paint brush if needed, but not after the male was within 5 cm from the female. The time taken for the male to approach the female varied between and within species; we therefore allowed 4 h from the introduction of the male before terminating an unsuccessful mating trial. We substituted the male if he repeatedly failed to respond to the paint brush stimulus.

We recorded the copulation duration using a stopwatch and noted which palp (left or right) was used first for each insertion (palp choice). Copulation duration refers to the entire time the tip of the embolus remained inserted in the epignal tract, with or without the male (in the case of remote copulation[37]). Full palp breakage only occurred in *N. malabarensis* (Supplementary Figs. 7 and 8); those severed palps were immediately removed (Supplementary Fig. 7) to prevent continuous sperm transfer. Although full emasculation did not occur in the other four species, the males of those species do occasionally engage in lesser genital mutilation, and when they do, a broken embolus may form a mating plug lodged in the female spermatheca (Supplementary Fig. 9)[18,19,37].

We also recorded whether the male or the female terminated the copulation. If the former occurred, the trial was categorised as an insertion (or palp breakage in the case of *N. malabarensis*). If the female terminated copulation, it was either an attempt to cannibalise the male; or a true cannibalism of the male (Supplementary Fig. 10). Thus, the female was classified as cannibalistic or not, and the male was classified as being cannibalised or not. We then calculated the sexual cannibalistic rates of the spiders, and compared them to the literature reports[15,17,24,44–46,48]. In all species, male usually inserted one of its palp in one genital opening of female, and once we observed that a male inserted one of its palp consecutively in both genital openings of females, we discarded that pair in data analysis.

All trials were recorded using a digital HD camcorder (JVC GZ–MG50AG and JVC GX–MG50AG, Japan). Subsequent video playback allowed us to verify the measured copulation duration and observed palp choice.

**Post-copulation procedures.** The following four scenarios occurred during and post-copulation (Supplementary Fig. 10): (1) If the male initiated the end of a successful copulation (i.e., after insertion), we separated the male and the female. (2) If the female attempted to cannibalise the male or displayed aggressive attacks, such as kicking to displace the male from her epigynum during copulation, we immediately forced the aggressive female to retreat to a corner using a wooden stick. We then recorded any broken body parts of the male. (3) If the female cannibalised the male before the start of copulation, we immediately removed the male. We did not reuse the female for any future mating trials to ensure that our sample remained unbiased. (4) If the female cannibalised the male during copulation, we immediately separately the male from the female to prevent his further body damage as the palps were required intact for sperm counting. After these steps, we euthanised the spiders with carbon dioxide ($CO_2$) for 20 min.

**Female sexual cannibalism.** Sexual cannibalism is broadly defined as any female hostile behaviour toward males, which includes displaying aggressiveness, attacking, grabbing, or killing the male. Aggressiveness is the mildest extent of sexual cannibalism, such as showing signs of reluctance to mate. Attacking refers to the female giving chase or attempting but failing to grab the male. Grabbing occurs when the female moves to catch and hold on to the male, prior to biting him. We define killing as when the female bites and injects her venom into her partner. Although sexual cannibalism in nature often culminates in the actual consumption of the victim, our definition of cannibalism needed to stop with the killing itself (and not consumption) because, as per the experimental design described above, we separated the victim from its aggressor before his likely consumption. Therefore, within this study, we use a restricted definition of sexual cannibalism as females that grabbed and/or killed their mates.

**Palp detachment procedures.** If full-emasculation occurred, we removed the broken palp from the female epigynum immediately. We also removed the unused palp from the male body using soft forceps and scalpel under a stereomicroscope (Supplementary Fig. 11); this procedure ensured the severance of the intact palps between the palp tarsus and tibia, i.e., at the position of natural emasculation[62]. We stored all palps in 0.2 ml Eppendorf tubes and refrigerated them for sperm counting.

**Epigynum dissection and sperm counting procedures.** The epigynum refers to a sclerotized plate on female abdomen containing genital openings on the surface and spermathecae and ducts in its interior[63]. To determine if and how much sperm were transferred to the female genital tract, we dissected the epigynum in its entirety, then cleaned the soft tissue covering internal genitalia using soft forceps, scalpel, and a 27G × ½ inch fine injection needle (BD PrecisionGlide, USA) under a

stereomicroscope. This procedure exposed the undamaged spermathecae, copulatory ducts and fertilisation ducts (Supplementary Fig. 12); not damaging the spermathecae was important to prevent sperm loss, particularly in *L. decorata* with fragile inner organs. We individually stored the isolated female epigyna in 0.2 ml Eppendorf tubes and refrigerated them before sperm counting.

**Sperm counting procedures**. We counted sperm in the used and unused palps, as well as the epigynum. Since females have paired spermathecae, we counted the sperm in the spermatheca that was used during copulation with a particular palp. The number of sperm in the used palp and spermatheca gave the total number of sperm in the palp chosen by the male for copulation (Supplementary Fig. 13). Sperm clumping is an issue that complicates sperm quantification as it often underestimates the total sperm count, causing inaccuracy and unreliability of the sperm numbers. To ensure that this problem is minimised, the sperm counting procedure was modified by referring to multiple references[1,13,28,64] and using 'mix-and-match' to derive a best method. Sperm counting was conducted separately for the used palp, unused palp and epigynum. Since females have paired spermatheca, the sperm count was taken from the one spermatheca that was used with a particular palp. The number of sperm in the used palp and the spermatheca gave the total number of sperm in the palp the male has chosen to use for copulation (Supplementary Fig. 14).

We modified the sperm counting procedure from earlier studies[37,43,65,66]. We added 5 µl of 0.9% saline + 10% triton-X detergent to the Eppendorf tube containing either a palp or spermatheca and used metal forceps to thoroughly grind and break the reproductive organs for sperm to be released into the solution. We then added another 5 µl of solution to wash the forceps of any debris and sperm which might have accumulated. Samples containing the palp were ultrasonificated (Microcomputer heater–timer controller DSC–120TH, Sonicor Instrument Corporation, USA) five times 30 s after which the Eppendorf tube was manually flipped to the other side so that the solution was evenly sonicated to break up sperm clumps. The samples then underwent another 30 s of pulse-vortexing (Type 37600 Mixer, Thermolyne Corporation, USA). Samples containing the spermatheca underwent a similar procedure, except that the period of ultrasonification was extended to five min (sonicated ten times 30 s) as sperm clumping (Supplementary Fig. 14) was observed to occur more frequently.

We extracted two 4 µl samples using a pipette and placed them on both chambers of a 0.1 mm Neubauer improved haemocytometer (Blaubrand, Wertheim, Germany). Using a compound microscope at 400× magnification, we counted sperm in all 25 quadrats (0.05 × 0.05 mm) from each chamber. As only encapsulated sperm could be observed under the microscope, we equated the number of capsules to the number of sperm[37]. An average was calculated from the quantities in the two chambers, which was then multiplied by the dilution factor to get the total number of sperm of the sample. We also calculated the percentage of sperm transferred to the female during the copulation using the formula from Snow and Andrade (2004)[65]:

$$P = [T/(U+T)] \times 100\% \qquad (1)$$

Where $P$ is the percentage of sperm transferred to female's spermatheca, $T$ is the total number of sperm in the female's spermatheca, and $U$ is the total number of sperm counted in the used palp.

**Statistics and reproducibility: comparative study**. All statistical analyses were done using R 4.1.1[67]. We checked for data normality using the Shapiro–Wilk tests and transformed the data if it were not normally distributed. We used parametric or nonparametric tests depending on the normality of the data.

We used two GLM with a Gamma error and log-link function to test for the effects of sexual cannibalism (whether a female is cannibalistic) and other five exploring factors (sexual dimorphism [the ratio of female mass to male mass], male post-maturation age, female post-maturation age, species, and taxonomy [family to which a species belongs to]) on the two mating strategies (palp used with more sperm and percentage of sperm transfer) separately. For all GLMs, we used an information-theoretical framework[68] to assess a series of hypothesis-driven models. We then compared the models using Akaike's information criterion corrected for small sample sizes (AICc) to identify the best fitting model that predicts the effects of explorative factors on each male mating strategy[68] by ranking them based on the AICc values using *model.sel*, and averaged the models using *model.avg*, if ΔAICc < 2, in the R package MuMIn[69].

We coded the ratio of the number of sperm in the used and unused palp and percentage of transferred sperm as response variables. In these two GLM models, the six main factors and their interactions were predictors, but we included copulation duration as additional predictor when testing for the effects on the rate of sperm transferred to control for its effect. We excluded male and female size from the model because body size measurements are unrelated to sperm transfer[70]. Instead, we used their mass ratio as SSD. Regression estimations were also conducted on the males of the five individual species to test for a correlation between male carapace width and the total number of sperm it had. The correlations were very weak in all five species, with the highest correlation in *H. multipuncta* ($R^2 = 0.470$, $F_{1,9} = 7.984$, $N = 11$, $P = 0.02$). Therefore, we excluded male size from the models.

All five species differed greatly in copulation duration and percentage of sperm transfer (see "Results"), which might reflect species specific behaviours. We ran a GLM for each individual species to identify factors which might have species specific effects, but the GLM was not feasible due to a small sample size (smallest sample in *H. multipuncta* = 11). As this study focused on sexual cannibalism as the hypothetical driving factor behind the three male mating strategies, combining species into groups may not be the best solution to solve the problem of small sample sizes because of the dilution of cannibalism effects. We thus did not report the results from tests for individual species here.

**Hypotheses and predictions for the palp choice mechanisms**. The following elaborates how the formulated hypotheses may form the basis underlying palp choice during copulation. The hypotheses are grouped according to two ways a male spider may fill up its palps with sperm. First, mechanism A assumes that a random amount of sperm may be loaded into each palp and the spider is then able to detect minute differences in sperm quantity of each palp to make a choice of palp during copulation. Second, mechanism B assumes that the male may intentionally load one palp with more sperm than the other and thus prefer to use the palp containing more sperm. In this latter case, memory may be involved in palp selection.

**Mechanism A**. (1) Volume detection. The male spider is able to detect differences in amount of sperm in each palp and thus prefers to choose the palp with more sperm for copulation. The male is either able to detect the absolute difference in the sperm amount or it is unable to detect the absolute difference in sperm amount, and instead detects the ratio of sperm amount or proportional difference. The latter refers to the proportional processing of information in decision making, where the traits are judged as a relative to the presented stimuli. This cognitive mechanism is magnitude-dependent, where at high magnitudes of a trait value a larger increase is needed to obtain the same proportional difference than at low magnitudes[38,39,42]. This could have evolutionary consequences, that when a trait is at a high magnitude it will require more investment to exaggerate it further to achieve a proportional difference sufficient for detection. Therefore, due to cognitive constraints, males may select the palp containing more sperm not by evaluating the absolute, but instead, the proportional difference or the ratio of sperm amount between the two palps. (2) Mass of palp. The male spider may select the palp with greater mass for copulation. The male spiders of sexually dimorphic and cannibalistic species are suspected to be sensitive to differences in mass; studies have shown a change in behaviour of male spiders subsequent to palp severance, thus subsequent to mass loss[17,60]. The mass of a palp may be linked to the volume of sperm it contains, or to palp dominance or palp capacity. However, the mass of palp is difficult to quantify accurately. (3) Size and capacity of palp. The male spider may have a tendency to select the palp with a larger palpal bulb (part of the palp where sperm is stored) for copulation. A larger palpal bulb is possibly positively correlated to a higher capacity for sperm and this may account for the tendency to select the palp with more sperm for copulation. Size and capacity of palp may be linked to mass of palp. However, size and capacity of palp is difficult to quantify accurately, especially because palps are damaged after emasculation. (4) Palp dominance (subconscious). A dominant palp may naturally tend to draw up more sperm during sperm induction. The male spider may then subconsciously and consistently use the dominant palp for copulation regardless of most circumstances.

**Mechanism B**. (1) First palp preference. A male spider conducts sperm induction by spinning a sperm web, placing a drop of sperm on the web from its genital pore on the abdominal venter, then filling its palps with sperm. It was implied that male spiders may fill up their palps one at a time[71]. We have made an assumption that the male spider may place a limited amount of sperm on its sperm web during sperm induction. Therefore, the palp that the male spider loads with sperm first has the potential to be filled to its maximum capacity, while the second palp can only take up the remaining sperm on the sperm web. The male may then remember to use the first palp for copulation later on. The preference for first palp may be linked to palp dominance, whereby the first palp is the dominant palp. (2) Palp dominance (conscious). The male spider may intentionally fill its dominant palp with more sperm and memory may or may not be involved in selecting the palp containing more sperm.

**Study species**. For this experiment, we focused on *N. malabarensis*, a highly sexually size dimorphic nephilid/araneid[29,62] that is common in Singapore and easily recognised in juvenile stages due to its highly characteristic web built on trees (Supplementary Fig. 9)[72]. We collected juveniles and subadults in Singapore and reared them to maturity (for rearing details, see the "comparative study" section). Only virgin spiders were used in the mating trials.

**Experimental procedures**. We followed the procedures as presented in Supplementary Fig. 15, including mating trial 1 and subsequent mating trials up to 5. If the male broke its palp, the male was not used in the subsequent mating trials, and the sperm of the used palp for its first insertion and the unused palp were counted. If the male did not break its palp, it was used for the subsequent trials until it broke

one of its palps, and the sperm of its palps were counted. For detailed procedures, see the descriptions below.

**First mating trial**. Newly matured virgin males (mean ± s.e.m. post-maturation age: 39.5 ± 5.1 days) were used in the first mating trials. The first mating trial is the first step to changing the quantity of sperm in one of the male's palps, such that the male's palp choice in subsequent mating trials can give an indication of the basis underlying palp choice.

We randomly selected a virgin male and a virgin female for a mating trial. We placed the male at least 10 cm from the female on the female's web to allow it to proceed with courtship and copulation, whereby the embolus of the palp is inserted into the epigynum. We recorded which palp the male used for copulation, the copulation duration and whether the male severed its palp(s) with a HD video camcorder (HDR-SR8E, SONY, Japan). If copulation failed to occur within 5 h of the start of the mating trial, the male and female were separated and left to rest for a day before being experimented upon again.

After a successful mating trial, we euthanised the female with $CO_2$ for 20 min to record its body mass and size, and dissect its spermathecae (Supplementary Fig. 12d) for sperm counting. An experimental eunuch male was euthanized similarly, but a male with intact palps after a mating trial was reused. The size and mass of euthanized males were measured and their palp(s) were stored for sperm counting.

**Sperm counting 1**. After the first mating trial, we counted the amount of sperm in the spermatheca of the mated female and the palps of the male that had lost at least one palp. The experimental procedure for sperm counting was as in the comparative study above.

**Subsequent mating trials**. Following a successful first mating trial, we promptly sealed the gonopore of experimental males with intact palps with acrylic paint to prevent further sperm induction. This experimental modification is a precautionary measure to block the genital pore, which males use to release sperm from their testes onto a sperm web and then into the palps. The male was then left to rest for at least 15 min to allow the paint to dry before being subjected to the second mating trial with a different virgin female. The mating trial procedures were as in the first mating trial. We recorded the palp the male used for copulation, whether the palp was severed, and the duration of copulation.

Each male underwent a maximum of five mating trials, each with a different female, or until the male finally severed his palp(s) or was killed. The multiple mating trials aimed to change the amount of sperm in the palps of the male after each successful copulation with a different female. After mating, we euthanised the female, measured her body mass and size, and dissected her epigynum for sperm counting. We also euthanised the males that underwent the mating trials and cut the non-severed palps using a scalpel for sperm counting. The males that refused to mate after five trials were likewise euthanised, dissected and measured.

**Sperm counting 2**. Previously, we counted the sperm from the spermathecae of females that had successfully mated, and of males that had not severed their palps. In addition, we counted the sperm within the male's palp only after the male had severed at least one palp and was no longer able to exhibit palp choice in mating trials. Here, sperm counting was conducted for both the palps of the remaining males and in the spermathecae of the last female it mated with. The males' palps were dissected and the same experimental procedures were followed, except that the solution containing sperm from palps were ultrasonicated for 30 s for five times only.

**Statistics and reproducibility for the data on palp choice mechanisms**. The amount of sperm found in each female and the male's palps can be used to deduce the amount of sperm present in each palp before each mating trial. Therefore, the amount of sperm that the male holds can be tracked. We performed all analyses in R. Checking for data normality, data transformation, as well as the use of parametric or nonparametric tests, were done as above in the comparative study.

Results from our comparative study showed that significantly more males used the palp with more sperm for the first insertion. Here, we used a paired samples t-test and Chi-square test for goodness-of-fit to determine if males generally selected the palp with more sperm during the first copulation. We then used Wilcoxon signed rank test to test if there was a significant difference in amount of sperm between the used versus not used palp in subsequent mating trials (in those males that mated more than once). We also used Chi-square test for goodness-of-fit to analyse whether males in subsequent mating trials had a preference for using the palp with more sperm for the first and subsequent insertions.

We performed Chi-square test for goodness-of-fit to test whether males had a tendency to change their choice of palp in subsequent mating trials. The preference for consistently using the same palp in all mating trials would support the hypothesis that palp dominance influences palp choice. We also performed binary logistic regression to assess the impact of alternation of palp on the ability to choose the palp containing more sperm. We included the male identity as a random effect in the model because individuals were used repeatedly. The Chi-square test for goodness-of-fit and binary logistic regression model only applied to males that did not sever their palp in the first mating trial because males that had lost at least one palp in their first mating trial were unable to show alternation of palps in subsequent trials.

To test the palp choice based on volume, we used the mixed-effects model with binomial error and logit-link function using the R package *lme4*[73] to assess the impact of several factors on the ability of a male spider to choose the palp containing more sperm. We included male identity as a random effect, and included positive difference in the amount of sperm between the two palps of a male, sperm ratio (i.e., the ratio of the amount of sperm present in the palp with more sperm to that in the palp with less sperm), SSD (the ratio of female to male mass) and emasculation frequency as fixed effects. The positive difference and sperm ratio showed the magnitude of the difference between the quantity of sperm present in each palp and did not take into account whether the male had used the palp for copulation. Positive difference and sperm ratio were derived from the amount of sperm present in both palps before each mating trial, detected through sperm counting. As described above in the comparative study, we also used an information-theoretical framework to assess a series of hypothesis-driven models[68]. We then compared the models using AICc to identify the best fitting model that predicts the effects of explorative factors on palp choice. We identified the best by ranking them based on the AICc values using *model.sel*, and averaged the models using *model.avg*, if ΔAICc < 2, in the R package MuMIn[69].

As no factors in the models had significant effects on palp choice (see "Results"), we checked the values of the positive difference and sperm ratio in relation to the ability to choose the palp containing more sperm (for all mating trials). According to proportional processing, we assumed that a low positive difference or a low sperm ratio might result in random palp choice, since males might not be able to distinguish between small differences. However, a large positive difference or a high sperm count ratio resulted in choosing the palp with more sperm. We manually set the 75th percentile of the boxplots as a cut-off point to distinguish high and low values of positive difference and sperm count ratio. We then performed four Chi-square tests for goodness-of-fit to test for a significant difference in the occurrence of high and low (according to the set cutoff) positive differences and sperm count ratios, depending on the male choice of palp.

**Reporting summary**. Further information on research design is available in the Nature Research Reporting Summary linked to this article.

## Data availability

All data that support the findings of this study required to reproduce the analyses of the data are available in Supplementary Data 1 and in the Zenodo repository[74], https://doi.org/10.5281/zenodo.6434021.

## Code availability

All codes required to reproduce the analyses of the data are available in the Zenodo repository[74], https://doi.org/10.5281/zenodo.6434021.

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

## Acknowledgements

We thank Poh Moi Goh and Tommy Tan for their laboratory assistance, as well as Qiqi Lee, Joelyn Oh, and Hui Ai for their help in the laboratory experiments. We also thank Jutta Schneider and an anonymous reviewer for their valuable reviews of this manuscript. Spiders used in this research were collected under the permit (NP/RP12-037) issued by the Singapore National Parks Board. The research was supported by the Ministry of Education Academic Research Fund (AcRF) grants (A-0004443-00-00) and the National Natural Science Foundation of China (31801979 and 31872229). M.K. was partially supported by the Slovenian Research Agency (J1-9163, P1-0255).

## Author contributions

D.L., S.C.Z., M.K., N.Y.L.T. and X.X.B.W. designed the study. N.Y.L.T., X.X.B.W., and S.C.Z. conducted the experiments. D.L., S.C.Z., N.Y.L.T., X.X.B.W., and L.Y. analysed the data. D.L., S.C.Z., N.Y.L.T., X.X.B.W., M.T., and M.K. drafted the manuscript. All authors contributed to data interpretation and revision of the manuscript.

## Competing interests

The authors declare no competing interests.
