## [Peer Review File · Communications Biology]

Reviewers' comments:

Reviewer #1 (Remarks to the Author):

Sexual cannibalism is particularly common among spiders and represents a regular component of mating in species that are characterised by low male mating rates. In such species, males are specialised to increase their paternity by reducing sperm competition. Female cannibalism can be more or less costly for the males; as the cost depend on if and how strongly it affects males' optimal mating rates. If sexual cannibalism imposes fitness costs on the males, antagonistic traits will be favoured by selection. The present article postulates three potential male counter adaptations to sexual cannibalism using five species from 5 orb-weaver genera and 2 families.

The authors pose four hypotheses:

the "better charged palp" hypothesis predicts that males preferentially use the pedipalp that contains more sperm for their first copulation;

the "prolonged copulation" hypothesis and the "in-and-out" hypothesis make opposite predictions as to how a male should respond to the threat of sexual cannibalism

the "fast sperm transfer" hypothesis predicts that cannibalism should select for increased sperm transfer rates.

The authors found evidence to support the better charged palp and some indication for effects other than sexual cannibalism on copulation duration.

While hypotheses 1 and 4 make straightforward testable predictions, hypothesis 2 and 3 are a bit vague and they appear very difficult to test with only a handful of species and rather low variation in sexual cannibalism but variation in many other traits. The duration of copulation varies a lot between species and also within species so that a limited taxon sampling might produce ambiguous patterns. It would be, in my opinion, sufficient to focus only on the rate of sperm transfer (either in relation to the numbers of sperm in the pedipalp as used here or as transfer speed or both) and compare related species with and without sexual cannibalism (or with large difference in the prevalence of sexual cannibalism to test predictions on sperm transfer rate.)

I must admit that I really struggled with this paper and the format of providing methods at the end plus relying heavily on supplementary material. I am also not an expert on complex GLM models with lots of variables and am easily confused when trying to interpret manyfold interactions and correlations. Especially comparing 5 distant species on the basis of relatively few mating trials per species appears to harbour a high risk of type II error. In contrast, I really liked the part on *N. malabarensis*.

INTRODUCTION

Line 67: I would argue that monogyny actually predates the adaptations you are listing and would be the cause for all the other adaptations as a consequence of male biased sex ratios.

Line 88: I am not aware of examples in orb-web spiders where sperm is expelled or removed

Lines around 100: I would argue that you don't need the longer or shorter copulation hypotheses if you focus on sperm transfer alone. This would remove some unnecessary complexity and would aid readability. How many sperm are transferred will be what ultimately matters for paternity and male fitness. Female attacks during copulation often end copulation and select for a faster transfer rates. Alternatively, males may find ways to prolong copulation despite female aggressiveness and these are the well-known tactics that involve self-sacrifice, mating while the female is feeding or moulting and ultimately remote copulation. However, you do not consider such strategies here at all but put all species into the same pot. Putting the argument in the above way seems more logical to me and you would end up with 2 instead of 4 hypotheses.

RESULTS:

SEXUAL CANNIBALISM (lines 121-131; 534; supp): the score (1-5) of sexual cannibalism is broadly defined and includes cannibalism or attempts before, during and after copulation. Even though all timings and outcomes are combined, the rates are comparatively low and do not differ to a large extent between the species. I also caution that only cannibalism during copulation would be a relevant selection pressure on copulation duration and sperm transfer rate. Since you strive to explain the effect of sexual cannibalism on sperm use and transfer, I do not see how these traits could be under selection by precopulatory female aggression. Given that you have a

very small number of species, one outlier has a large influence on your results. In the supplemental material you show two criteria and state that you used the stricter one that only include harmful behaviour which I appreciate. I understand that you lump attacks and kills occurred before, during and after copulation but I am not entirely sure if that is correct as it is not explicitly stated (or I overlooked it).

I am further concerned about the reliability of your estimates of sexual cannibalism as they are biased on (at least partly) rather few mating trials per species and the trials were not standardized for factors known to influence sexual cannibalism such as female age, size and weight. Your estimates appear to differ from rates in other studies (mentioned in supplement).

In the section on *N. pilipes* in the supplement you refer to one of my papers as evidence that cannibalistic females gained more mass between maturation and oviposition. It is not made very clear that this was on another species (*Trichonephila plumipes*) and that the effect was independent on actual cannibalism because it occurred even though females were prevented from consuming the males. It is not a good prove that females benefit from sexual cannibalism particularly not in a different species.

SPERM COUNTS (lines 132 ff): I find the section on sperm counts a bit confusing. No differences between sperm counts in the two palps of males were found but males more likely chose the one with more sperm, independently of the right or left side. This is a very interesting finding. (It is unfortunate that the result section is so truncated that I need to go to the supplement to find out which tests were used to compare the sperm numbers in the palps. I would recommend changing the figure to depict the paired data.) Then you compared the frequency of males that used the palp with more sperm. Did these only include the trials that found a significant difference between the used and unused palp? If the used palp had slightly more sperm than the unused palp, how did you classify such a case? This is particularly curious in *N. pilipes* where there is no significant difference between used and unused whereas the frequencies show that 8 of 10 used the palp with more sperm.

In the GLM (Table 1) you find that males were more likely to use the palp with more sperm if the female was relatively larger, old and cannibalistic. While it makes sense that males do that in cannibalistic species, where the chances of ever using the 2nd palp are small, the other results speak against the hypotheses that this behaviour is a response against sexual cannibalism. It seems better explained by male mate choice for older and larger females.

Lines 154 ff: I also do not quite understand this section. How did you select the variables to enter into the model? I would be concerned that some variables are correlated such as age and mass. Did you exclude this? Why did you use the mass ratio to estimate SSD? Females appeared to differ a lot (in *N. malabarensis* up to 6 weeks) in post-maturation age and I would expect that weights differ significantly between young and old females. Why did you not use a fixed body trait such as carapax width? Since you measured it anyway, I would recommend to use it instead of weight for getting a more realistic and comparable estimate of SSD.

Fig. 2: I am again confused how the variable "sexual cannibalism" entered this figure. Is it the score of each individual mating trial? Or the overall estimate for the species?

It is interesting that in 4 of the 5 species, SSD not only explains whether males use the better palp but it also correlates positively with copulation duration (similar findings exist for several other species). This is true for 4 species while the one species with remote copulation does not follow this pattern. This would be worth discussing in more detail. The evolution of remote copulation releases males from the pressure of sexual cannibalism and forceful dislocation of the pedipalps.

Line 224ff: The rate of sperm transfer is an important measure if assessing male paternity success. I am confused what you mean by rate. I would expect that rate means how many sperm are transferred per unit time of copulating. However, you computed the proportion of sperm transferred from the total amount of sperm that was in the pedipalp. Hence a male that copulated for longer and transferred a larger proportion of the sperm in their pedipalp had a higher rate?

What if you included copulation duration in the model? (or did you?)

Generally, I wonder how frequently males actually empty all sperm from a palp during copulation or put in other words, I wonder how relevant it is for paternity success how many sperm a pedipalp contained? If males would rarely empty their palps (as your data suggest), then only the duration of copulation and the rate of transfer per time would matter. These arguments are likely not valid if there is the option of remote copulation.

I question the use of male age as an explanatory variable. How relevant are age differences of a few days and why did you use age rather than size?

No information is provided about insemination patterns and palp-reuse. In many entelegyne

spiders each pedipalp can only inseminate one of the two spermathecae and in *Argiope* palps are only used once because they become dysfunctional after breakage for means of plugging. How is this in *N. malabarensis*? Can each pedipalp be used to mate into each genital opening? If not, the repeated use of the same pedipalp should depend on how much sperm was transferred during the first mating.

I am also surprised that you excluded size from the models based on evidence derived from a single and different spider than used here. Size of the female for example may well explain sperm transfer. Males can be selective and invest more in large than in small females.

General comments:

I apologize for the somewhat chaotic commenting. I find the topic very interesting as it is very close to my own research interest. However, it was very hard to comment on a paper that has so much content spread over several sections having to scroll back and forth. This does not work for me. I spent several days on that manuscript and still feel that I did not do it justice as I might have overlooked or confused information. After two extensions, I will hand in my review now although I am not satisfied with it.

In my opinion, the content and the frame work of this very interesting study is too complex for a single paper with space restrictions. However, I would really like to see these data published but perhaps in more than one publication.

In the present paper, I liked the better-charged-palp idea best and the experiments on *N. malabarensis* provide convincing support for the idea. Plus, the species is unique with the adaptation of remote copulation. I would put that single species study in the focus rather than the comparative approach.

Jutta Schneider

Reviewer #2 (Remarks to the Author):

This manuscript compares male mating strategies in systems with varying levels of sexual cannibalism. The authors assess 4 hypotheses (better charged palp, prolonged copulation, in-and-out and fast sperm transfer) that are each proposed to improve male reproductive success in the face of potential cannibalism. They also identify sperm volume as the mechanism by which males choose which palp to use. The experiments are elegant and it is nice to see the authors also identifying a mechanism by which their hypotheses are supported. Research into cannibalistic systems often focus on female adaptations, and so the results reported here will be of broad interest.

Minor comments:

- Figs S1-4 are at the end of the supplementary materials, making them difficult to locate. Suggest placing them in consecutive order.
- The phrase 'cognitively perform palp choice' is rather awkward (lines 39 & 116)
- '...we studied *N. malabarensis* through a series of follow-up experiments.' Can you phrase this using active voice i.e. we tested male genital choice...through a series of follow-up experiments.
- Lines 130-131: seem contradictory. It is unclear in this paragraph whether all percentages refer to cannibalism during copulation (as opposed to before, after, or all stages).
- There appears to be a slight difference in the descriptions of the fast-sperm transfer hypothesis between the manuscript and the supplementary material. The manuscript describes an increase in sperm transfer rate, the supplementary material (line 19) transferring more sperm.
- Table S2, Fig. 1 and lines 143-4: it is much easier to follow 'the palp with more sperm for their first insertion' than 'no. of males that used the palp with more or less sperm'
- Figure 1a: The stats on the figure suggest that there is no difference in sperm number between the left and right palp (contrast with figure legend).
- Line 209: Typo 'females'
- Figure 4: It is interesting that in general quite low percentages of sperm are being transferred. Any hypotheses?
- Line 260: How were genital pores sealed?

- Line 373: Suggest explaining 'higher ratio' more fully here – there are lots of results and it can be hard to remember what each means or is referring to
- Line 377: Add 'questions' after 'these'
- Lines 428-442: The description of Weber's law makes the rationale underlying the methods here much clearer. I suggest placing a short (1-2 sentences) outlining this into the results around Line 159, as reading the paper consecutively the 'ratio' doesn't make sense until much later.
- Line 459: What does 'extrapolated from the same genus' mean?
- Lines 474-6: Suggest deleting the final sentence, which doesn't add much, and incorporating the sentence in lines 474-5 into the end of the previous paragraph.
- Lines 553-554: How were broken palps removed immediately? Was this after euthanasia with CO2 for 20mins?

Reviewer #1

Sexual cannibalism is particularly common among spiders and represents a regular component of mating in species that are characterised by low male mating rates. In such species, males are specialised to increase their paternity by reducing sperm competition. Female cannibalism can be more or less costly for the males; as the cost depend on if and how strongly it affects males' optimal mating rates. If sexual cannibalism imposes fitness costs on the males, antagonistic traits will be favoured by selection. The present article postulates three potential male counter adaptations to sexual cannibalism using five species from 5 orb-weaver genera and 2 families.

The authors pose four hypotheses:

the “better charged palp” hypothesis predicts that males preferentially use the pedipalp that contains more sperm for their first copulation; the “prolonged copulation” hypothesis and the “in-and-out” hypothesis make opposite predictions as to how a male should respond to the threat of sexual cannibalism, the “fast sperm transfer” hypothesis predicts that cannibalism should select for increased sperm transfer rates.

The authors found evidence to support the better charged palp and some indication for effects other than sexual cannibalism on copulation duration.

While hypotheses 1 and 4 make straightforward testable predictions, hypothesis 2 and 3 are a bit vague and they appear very difficult to test with only a handful of species and rather low variation in sexual cannibalism but variation in many other traits. The duration of copulation varies a lot between species and also within species so that a limited taxon sampling might produce ambiguous patterns. It would be, in my opinion, sufficient to focus only on the rate of sperm transfer (either in relation to the numbers of sperm in the pedipalp as used here or as transfer speed or both) and compare related species with and without sexual cannibalism (or with large difference in the prevalence of sexual cannibalism to test predictions on sperm transfer rate.)

ANSWER: Thank you very much for your suggestions. We have removed all contents related to the original hypotheses 2 and 3 (the “prolonged copulation” hypothesis and the “in-and-out” hypothesis) (Introduction: Lines 84-95; Results: lines 226-259; Discussion: lines 427-450), as suggested. In this way, we have focused on original hypotheses 1 (the ‘better charged palp’ hypothesis) and 4 (the ‘fast sperm transfer’ hypothesis). Regarding the scoring of sexual cannibalism, we prefer to retain our original quantification, because as we see it, simple presence/absence coding of cannibalism for the species level would yield lower resolution, and would weaken the statistical power of this variable.

I must admit that I really struggled with this paper and the format of providing methods at the end plus relying heavily on supplementary material.

ANSWER: We are sorry about this, which was due to the journal format. In an

attempt to make our reporting more easily digestible, we have revised the supplementary material by incorporating its text in the Methods part of the manuscript, leaving only supplementary figures and tables in the revised supplementary material.

I am also not an expert on complex GLM models with lots of variables and am easily confused when trying to interpret manifold interactions and correlations. Especially comparing 5 distant species on the basis of relatively few mating trials per species appears to harbour a high risk of type II error. In contrast, I really liked the part on *N. malabarensis*.

ANSWER: For the variables and descriptions in the GLM models, we have tried to make it clearer. In addition, since the data on our original hypotheses 2 & 3 related to copulation duration have been completely removed (Introduction: Lines 84-95; Results: lines 226-259), there are only two significant interaction terms detected in GLM testing for the original hypothesis 1 (the ‘better charged palp’ hypothesis), and we have provided more detailed description on these interactions (Lines 187-191, lines 193-197, lines 203-206).

INTRODUCTION

Line 67: I would argue that monogyny actually predates the adaptations you are listing and would be the cause for all the other adaptations as a consequence of male biased sex ratios.

ANSWER: To accommodate this thought, we have deleted the word “monogyny” from the examples we listed (Line 67).

Line 88: I am not aware of examples in orb-web spiders where sperm is expelled or removed

ANSWER: After having removed the contents related to original hypotheses 2 & 3, we also deleted this paragraph (Introduction: Lines 84-95; Results: lines 226-259).

Lines around 100: I would argue that you don’t need the longer or shorter copulation hypotheses if you focus on sperm transfer alone. This would remove some unnecessary complexity and would aid readability. How many sperm are transferred will be what ultimately matters for paternity and male fitness. Female attacks during copulation often end copulation and select for a faster transfer rates. Alternatively, males may find ways to prolong copulation despite female aggressiveness and these are the well-known tactics that involve self-sacrifice, mating while the female is feeding or moulting and ultimately remote copulation. However, you do not consider such strategies here at all but put all species into the same pot. Putting the argument in

the above way seems more logical to me and you would end up with 2 instead of 4 hypotheses.

ANSWER: As advised, we have deleted the two copulation duration-related hypotheses (Lines 84-95, lines 226-259).

RESULTS:

SEXUAL CANNIBALISM (lines 121-131; 534; supp): the score (1-5) of sexual cannibalism is broadly defined and includes cannibalism or attempts before, during and after copulation. Even though all timings and outcomes are combined, the rates are comparatively low and do not differ to a large extent between the species. I also caution that only cannibalism during copulation would be a relevant selection pressure on copulation duration and sperm transfer rate. Since you strive to explain the effect of sexual cannibalism on sperm use and transfer, I do not see how these traits could be under selection by precopulatory female aggression. Given that you have a very small number of species, one outlier has a large influence on your results. In the supplemental material you show two criteria and state that you used the stricter one that only include harmful behaviour which I appreciate. I understand that you lump attacks and kills occurred before, during and after copulation but I am not entirely sure if that is correct as it is not explicitly stated (or I overlooked it).

ANSWER: In this study, we did not use the mean value of sexual cannibalism for each species in our statistical analysis. Instead, we considered a female as cannibalistic if it attacked males during mating. That is, for each female, we coded as 1 or 0 if it attacked a male or not, respectively. We agree that only cannibalism during copulation would be a selection pressure on copulation duration. To make the conclusion reliable, we have deleted all the contents from the hypotheses related to copulation duration. For the palp choice in this study, we think that pre-copulatory attacks would also be relevant, because it means that a female is aggressive, leaving limited time for the male to make choice.

I am further concerned about the reliability of your estimates of sexual cannibalism as they are biased on (at least partly) rather few mating trials per species and the trials were not standardized for factors known to influence sexual cannibalism such as female age, size and weight. Your estimates appear to differ from rates in other studies (mentioned in supplement).

ANSWER: We agree. As said above, we did not use the mean value of sexual cannibalism for each species in our statistical analysis. Instead, we considered a female as cannibalistic if it attacked males during mating (scores explained above). To control for the effects of female age, size and weight, we have included female age and weight (mass) in all the models, including the interaction terms between sexual cannibalism and female age as well as between sexual

cannibalism and female/male mass ratio. See Supplementary Tables 2 and 3 in the supplementary material.

In the section on *N. pilipes* in the supplement you refer to one of my papers as evidence that cannibalistic females gained more mass between maturation and oviposition. It is not made very clear that this was on another species (*Trichonephila plumipes*) and that the effect was independent on actual cannibalism because it occurred even though females were prevented from consuming the males. It is not a good prove that females benefit from sexual cannibalism particularly not in a different species.

ANSWER: Thank you for pointing out our error. We have deleted the example as suggested.

SPERM COUNTS (lines 132 ff): I find the section on sperm counts a bit confusing. No differences between sperm counts in the two palps of males were found but males more likely chose the one with more sperm, independently of the right or left side. This is a very interesting finding. (It is unfortunate that the result section is so truncated that I need to go to the supplement to find out which tests were used to compare the sperm numbers in the palps. I would recommend changing the figure to depict the paired data.) Then you compared the frequency of males that used the palp with more sperm. Did these only include the trials that found a significant difference between the used and unused palp? If the used palp had slightly more sperm than the unused palp, how did you classify such a case? This is particularly curious in *N. pilipes* where there is no significant difference between used and unused whereas the frequencies show that 8 of 10 used the palp with more sperm.

ANSWER: We found no significant difference in sperm amount between the left and right palp (Fig 1a, Table 1 in the revised version), but detected a significant difference in the used and unused palp (Fig. 1b, Table 1 in the revised version). We did not exclude any trials. If the male used the palp with more sperm for the first insertion, we classified it as having chosen the palp with more sperm, and vice versa. We have revised the methods part by combining the Methods part and the supplementary part, which should make the rewrite clearer.

In the GLM (Table 1) you find that males were more likely to use the palp with more sperm if the female was relatively larger, old and cannibalistic. While it makes sense that males do that in cannibalistic species, where the chances of ever using the 2nd palp are small, the other results speak against the hypotheses that this behaviour is a response against sexual cannibalism. It seems better explained by male mate choice for older and larger females.

ANSWER: Our results suggest that female aggressiveness positively correlates with female age, albeit weakly. We interpret this to mean that males are more

likely to use the palp with more sperm if the female is relatively old and cannibalistic, while female body size (and male size, as well as their ratio) has no significant effect (Table 2 in the revised version).

Lines 154 ff: I also do not quite understand this section. How did you select the variables to enter into the model? I would be concerned that some variables are correlated such as age and mass. Did you exclude this? Why did you use the mass ratio to estimate SSD? Females appeared to differ a lot (in *N. malabarensis* up to 6 weeks) in post-maturation age and I would expect that weights differ significantly between young and old females. Why did you not use a fixed body trait such as carapax width? Since you measured it anyway, I would recommend to use it instead of weight for getting a more realistic and comparable estimate of SSD.

ANSWER: We employed the *vif* function in R to check for multilinearity among female age, male age, female body size (carapace width), male body size, female mass and male mass, and detected no multilinearity problem. We used the mass ratio to estimate SSD because adult females may continue to grow in mass but not size, and thus using the wet weight (mass) just before the mating trials allows us to estimate SSD more accurately than carapace width can. However, since many studies do use linear size measures to quantify SSD, and to address this concern, we made additional analyses by using the ratio of carapace width to estimate SSD. The results of these additional analyses using GLMs (available upon request) resembled our original ones. Thus, we continue to report the results of our original estimation of SSD.

Fig. 2: I am again confused how the variable “sexual cannibalism” entered this figure. Is it the score of each individual mating trial? Or the overall estimate for the species? It is interesting that in 4 of the 5 species, SSD not only explains whether males use the better palp but it also correlates positively with copulation duration (similar findings exist for several other species). This is true for 4 species while the one species with remote copulation does not follow this pattern. This would be worth discussing in more detail. The evolution of remote copulation releases males from the pressure of sexual cannibalism and forceful dislocation of the pedipalps.

Answer: As explained above, in the figure, ‘cannibalistic’ and ‘non-cannibalistic’ denote the aggressiveness status of each female in each species. If the female grabbed, attacked, or killed the male, we coded her as cannibalistic (yes or 1), and if the female did not attack the male, she was coded as non-cannibalistic (no or 0). We have deleted the part about the relationship between SSD, palp choice and copulation duration, in order to narrow down our focus on only two hypotheses, as suggested.

Line 224ff: The rate of sperm transfer is an important measure if assessing male paternity success. I am confused what you mean by rate. I would expect that rate

means how many sperm are transferred per unit time of copulating. However, you computed the proportion of sperm transferred from the total amount of sperm that was in the pedipalp. Hence a male that copulated for longer and transferred a larger proportion of the sperm in their pedipalp had a higher rate? What if you included copulation duration in the model? (or did you?)

ANSWER: Thank you for pointing this out. We have changed the rate to mean the percentage of sperm transfer. As described in the Methods, we calculated the percentage of sperm transferred to the female during copulation using the formula from Snow and Andrade (2004)⁶⁵, i.e., % sperm transfer = the number of sperm transferred / total number of sperm in the used palp x 100%. Yes, we included copulation duration in the model to control its effect on sperm transfer, but the results from the model showed no significant effect of copulation duration on the percentage of sperm transferred ($\beta = 0.00003$, $t = 0.04$, $p = 0.97$; Table 3).

Generally, I wonder how frequently males actually empty all sperm from a palp during copulation or put in other words, I wonder how relevant it is for paternity success how many sperm a pedipalp contained? If males would rarely empty their palps (as your data suggest), then only the duration of copulation and the rate of transfer per time would matter. These arguments are likely not valid if there is the option of remote copulation.

ANSWER: Interesting point! We do not know how frequently males empty the palps during copulation. Male spiders employ various strategies to circumvent female sexual cannibalism, and to mate repeatedly. Thus, a male may not invest all sperm at any one time. In our study, we did not allow males to recharge their palps. In *Nephilengys malabarensis*, the relationship between paternity and copulation duration is currently unknown.

I question the use of male age as an explanatory variable. How relevant are age differences of a few days and why did you use age rather than size?

ANSWER: Because we were studying the sperm transfer of the male, and the sperm quantity and quality may be affected by spider age. In addition, the degree of female aggressiveness may be correlated with ages. Thus, we used spider age as explanatory variables. As said above, we did use SSD (ratio of mass) as an exploratory variable when testing both hypotheses.

No information is provided about insemination patterns and palp-reuse. In many entelegyne spiders each pedipalp can only inseminate one of the two spermathecae and in *Argiope* palps are only used once because they become dysfunctional after breakage for means of plugging. How is this in *N. malabarensis*? Can each pedipalp be used to mate into each genital opening? If not, the repeated use of the same

pedipalp should depend on how much sperm was transferred during the first mating.

ANSWER: To reliably count the sperm in each palp, we did not allow the tested males to copulate more than once. All males of all species in our study were thus allowed to only use one palp to insert into one genital opening. This manipulative experiment does not necessarily match the reality in nature, but was essential for our tests. Data from nature suggest that male *N. malabarensis* usually insert one palp into one genital opening during the short copulation.

I am also surprised that you excluded size from the models based on evidence derived from a single and different spider than used here. Size of the female for example may well explain sperm transfer. Males can be selective and invest more in large than in small females.

ANSWER: This issue is explained above, as is the additional analysis that used size, in addition to mass.

General comments:

I apologize for the somewhat chaotic commenting. I find the topic very interesting as it is very close to my own research interest. However, it was very hard to comment on a paper that has so much content spread over several sections having to scroll back and forth. This does not work for me. I spent several days on that manuscript and still feel that I did not do it justice as I might have overlooked or confused information. After two extensions, I will hand in my review now although I am not satisfied with it.

In my opinion, the content and the framework of this very interesting study is too complex for a single paper with space restrictions. However, I would really like to see these data published but perhaps in more than one publication.

In the present paper, I liked the better-charged-palp idea best and the experiments on *N. malabarensis* provide convincing support for the idea. Plus, the species is unique with the adaptation of remote copulation. I would put that single species study in the focus rather than the comparative approach.

Jutta Schneider

ANSWER: Thank you very much for the suggestion. We originally wanted to pack a lot of data and many analyses into one, solid, comprehensive paper rather than producing several small papers because we felt that this would offer a more solid test of the male mating syndrome. Your comments and concerns made us aware of the problem that the original manuscript was too dense to be easily read. We discussed this problem thoroughly and decided that while we would prefer to publish the data on all five species rather than a single one, we would simplify the analyses and the narrative to only cover two of the four original hypotheses. We hope that our newly rewritten paper will be seen as a good step

towards this study being accepted for publication.

Reviewer #2

This manuscript compares male mating strategies in systems with varying levels of sexual cannibalism. The authors assess 4 hypotheses (better charged palp, prolonged copulation, in-and-out and fast sperm transfer) that are each proposed to improve male reproductive success in the face of potential cannibalism. They also identify sperm volume as the mechanism by which males choose which palp to use. The experiments are elegant and it is nice to see the authors also identifying a mechanism by which their hypotheses are supported. Research into cannibalistic systems often focus on female adaptations, and so the results reported here will be of broad interest.

ANSWER: Thank you very much!

Minor comments:

- Figs S1-4 are at the end of the supplementary materials, making them difficult to locate. Suggest placing them in consecutive order.

ANSWER: We have re-ordered them as suggested.

- The phrase 'cognitively perform palp choice' is rather awkward (lines 39 & 116)

ANSWER: We have deleted the word 'cognitively' (Line 39, line 118, line 293).

- '...we studied *N. malabarensis* through a series of follow-up experiments.' Can you phrase this using active voice i.e. we tested male genital choice....through a series of follow-up experiments.

ANSWER: We have rephrased the sentence as suggested (Lines 125-127)

- Lines 130-131: seem contradictory. It is unclear in this paragraph whether all percentages refer to cannibalism during copulation (as opposed to before, after, or all stages).

ANSWER: We have added 'the process of first' between 'during' and 'copulation' as advised (Line 139).

- There appears to be a slight difference in the descriptions of the fast-sperm transfer hypothesis between the manuscript and the supplementary material. The manuscript describes an increase in sperm transfer rate, the supplementary material (line 19) transferring more sperm.

ANSWER: We have combined the supplementary text with that in the methods part.

- Table S2, Fig. 1 and lines 143-4: it is much easier to follow ‘the palp with more sperm for their first insertion’ than ‘no. of males that used the palp with more or less sperm’

ANSWER: As suggested, we have rephrased (Supplementary Table 2, Fig. 1).

- Figure 1a: The stats on the figure suggest that there is no difference in sperm number between the left and right palp (contrast with figure legend).

Answer: We have amended as suggested.

- Line 209: Typo ‘females’

ANSWER: The whole paragraph has been deleted as suggested by Reviewer 1.

- Figure 4: It is interesting that in general quite low percentages of sperm are being transferred. Any hypotheses?

ANSWER: In our study, we only recorded the first copulations, so the percentage of sperm transferred is relatively low. In nature, males may circumvent cannibalism for additional copulations.

- Line 260: How were genital pores sealed?

ANSWER: We have added text to explain (Lines 301-302).

- Line 373: Suggest explaining ‘higher ratio’ more fully here – there are lots of results and it can be hard to remember what each means or is referring to

ANSWER: We have added text to explain it (Line 422).

- Line 377: Add ‘questions’ after ‘these’

ANSWER: Added as suggested (Line 426).

- Lines 428-442: The description of Weber’s law makes the rationale underlying the methods here much clearer. I suggest placing a short (1-2 sentences) outlining this into the results around Line 159, as reading the paper consecutively the ‘ratio’ doesn’t make sense until much later.

ANSWER: We have added a sentence to describe Weber’s law (Line 188-191)

- Line 459: What does 'extrapolated from the same genus' mean?

ANSWER: It means that the reported species may not match the species we studied, but since they belong to the same genus, we compared the frequencies of sexual cannibalism from the genus level. We have revised the sentence (Lines 507-508).

- Lines 474-6: Suggest deleting the final sentence, which doesn't add much, and incorporating the sentence in lines 474-5 into the end of the previous paragraph.

ANSWER: Revised as suggested (Lines 523-524).

- Lines 553-554: How were broken palps removed immediately? Was this after euthanasia with CO₂ for 20mins?

ANSWER: Yes, the male was separated from the female immediately, and the palps were removed after euthanasia with CO₂ for 20 mins. We have explained in the text (Lines 705-707).

REVIEWERS' COMMENTS:

Reviewer #1 (Remarks to the Author):

The authors attended to all my comments and I found the manuscript much easier to comprehend now. I have only one small comments regarding the sentence in lines 162-163. It appears that a very is missing and one would not say "sperms" as far as I know.